# Gameplay Filters: Robust Zero-Shot Safety through Adversarial Imagination

**Duy P. Nguyen** [1]     **Kai-Chieh Hsu**[*1]     **Wenhao Yu**[2]     **Jie Tan**[2]     **Jaime Fernández Fisac**[1]

[1]Princeton University, United States     [2]Google DeepMind, United States

{duyn,kaichieh,jfisac}@princeton.edu,   {magicmelon,jietan}@google.com

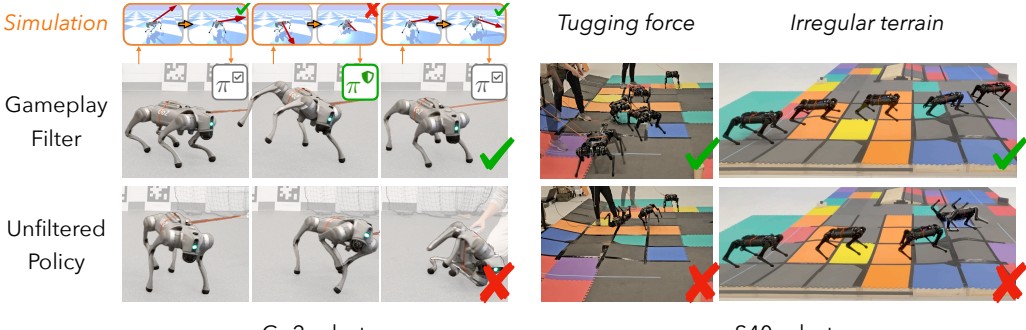

Figure 1: We demonstrate our game-theoretic predictive safety filter approach on two different quadruped robots equipped with safety-agnostic walking policies. The *gameplay filter* continually monitors the robot's safety by rapidly simulating worst-case futures invoked by a learned virtual adversary, and it preemptively disallows task-driven actions that would cause the robot to lose to it (violate safety). The gameplay-filtered robots exhibit rich and highly robust behaviors such as counterbalancing to fight persistent pulls and springing into a wide stance to break imminent falls.

**Abstract:** Despite the impressive recent advances in learning-based robot control, ensuring robustness to out-of-distribution conditions remains an open challenge. Safety filters can, in principle, keep arbitrary control policies from incurring catastrophic failures by overriding unsafe actions, but existing solutions for complex (e.g., legged) robot dynamics do not span the full motion envelope and instead rely on local, reduced-order models. These filters tend to overly restrict agility and can still fail when perturbed away from nominal conditions. This paper presents the *gameplay filter*, a new class of predictive safety filter that continually plays out hypothetical matches between its simulation-trained safety strategy and a virtual adversary co-trained to invoke worst-case events and sim-to-real error, and precludes actions that would cause failures down the line. We demonstrate the scalability and robustness of the approach with a first-of-its-kind full-order safety filter for (36-D) quadrupedal dynamics. Physical experiments on two different quadruped platforms demonstrate the superior zero-shot effectiveness of the gameplay filter under large perturbations such as tugging and unmodeled terrain. Experiment videos and open-source software are available online: https://saferobotics.org/research/gameplay-filter

**Keywords:** Robust Safety, Adversarial Reinforcement Learning, Game Theory

## 1   Introduction

Autonomous robots are increasingly required to operate reliably in uncertain conditions and quickly adapt to carry out a broad range of jobs on the fly [1–5]. Rather than synthesize an intrinsically safe control policy for every new assigned task, it is efficient to endow each robot with a *safety filter* that automatically precludes unsafe actions, relieving task policies of the burden of safety altogether.

8th Conference on Robot Learning (CoRL 2024), Munich, Germany.

Unfortunately, today's safety filter methods fall short of this promise for most modern-day robots. To cover a diverse range of tasks and environments, a safety filter needs to give the robot significant freedom to execute varied motions across its state space while robustly protecting it from catastrophic failures throughout this large envelope. To date, such *minimally restrictive* safety filters are only systematically computable for systems with 5–6 state variables [6–8], woefully short of the 12 needed to accurately model drone flight and the 30–50 needed for legged locomotion. Existing safety filters for high-order robot dynamics rely on reduced-order models [9–12]. These filters restrict the robot's motion to a local envelope, such as the vicinity of a stable walking gait, and become ineffective whenever the robot is perturbed away from it by external forces or unmodeled environment features (Fig. 1). How can we tractably and systematically compute safety filters that cover broad regions of robots' high-dimensional state spaces and a wide variety of deployment conditions?

**Contribution.** This paper introduces the *gameplay filter*, a novel type of predictive safety filter that can scale to full-order robot dynamics and enforce safety across a broad motion envelope and a designer-specified range of possible conditions (operational design domain). The filter is first synthesized by simulated self-play between a safety-seeking robot control policy and a virtual adversary that invokes worst-case realizations of uncertainty and modeling error (or *sim-to-real gap*). At runtime, the deployed filter continually rolls out hypothetical games between the two learned agents, overriding candidate actions that would result in the robot losing a future safety game. This methodology—based on the core game-theoretic principle that a strategy that wins against the worst-case opponent must also win against all others—unlocks real-time filtering in the robot's full state space by only requiring a single, highly informative trajectory rollout. We demonstrate the effectiveness of our approach experimentally on two quadruped robots that differ in physical parameters and sensing capabilities (Fig. 1). Each gameplay filter is synthesized and deployed using an off-the-shelf physics engine to simulate a manufacturer-provided robot model with a 36-D state space and a 12-D control space. We observe highly robust zero-shot safety-preserving behavior without incurring the conservativeness typical of robust predictive filters. To the best of our knowledge, this constitutes the first successful demonstration of a full-order safety filter on legged robot platforms.

**Related Work.** The last decade has seen important advances in robot safety filters. We briefly discuss the techniques most relevant to our work and direct interested readers to recent survey efforts [13–15] that shed light on safety filters' common structure and relative strengths.

*Value-based filters.* Hamilton–Jacobi (HJ) reachability methods use finite-difference dynamic programming to compute the best available safety fallback policy and the worst possible uncertainty realization from each state on a finite grid [6, 16, 17], which enables *minimally restrictive* safety filters. Although highly general, HJ computational tools suffer exponential blowup and do not scale beyond 5–6 state dimensions [18, 19]. Control barrier function (CBF) filters keep the system inside a smaller safe set while discouraging excessive control overrides [20]. CBFs lack a general constructive procedure and instead rely on manual design [21], sum-of-squares synthesis [22], or learning from demonstrations [23]. Robust formulations are comparatively less mature [24–27]. Self-supervised and reinforcement learning techniques can synthesize safety-oriented control policies and value functions ("safety critics") for systems beyond the reach of classical methods, but they are inherently approximate and offer no formal assurances [28–32]. Statistical generalization theory may be used to bound the probability of failure under the assumption that the robot can be tested on a statistically representative sample of environments and conditions before deployment [3].

*Rollout-based filters.* Predictive safety filters perform model-based runtime assurance by continually simulating—and in some cases optimizing—the robot's future safety efforts for a short lookahead time horizon [33–38]. Recent advances in fast forward-reachable set over-approximation [39–41] make it possible to check safety against all possible uncertainty realizations, although this runtime robustness comes at the cost of significant added conservativeness: for example, Hsu et al. [38] observe safety overrides 5 times as frequent as those of a least-restrictive HJ filter. Bastani and Li [35] instead propose sampling multiple possible trajectories, assuming a well-characterized disturbance distribution, to maintain a statistical guarantee. Our approach mimics Hsu et al. [38] in co-training a safety controller and a worst-case disturbance through simulated self-play, but it eschews over-conservative reachable sets by instead simulating a single closed-loop match between the two.

*Legged robot safety filters.* Legged robots have attracted increasing interest from researchers due to their versatility and increasing availability, as well as their challenging high-order and contact-rich dynamics [42]. Recent simulation-trained controllers leveraging domain randomization are showing promising agility and adaptability [1, 2, 43, 44]; however, robustness to out-of-distribution conditions cannot be easily quantified and remains an open issue. Unfortunately, all safety filters demonstrated on legged robots to date are based on simplified reduced-order dynamical models [3, 10–12], sometimes combined with local analysis around nominal walking gaits [9, 45, 46]. The dynamic envelope protected by these safety filters is limited to local state space regions where the simplified models apply, and their robustness to disturbances and modeling errors is contingent on the effectiveness of low-level tracking controllers. Our demonstration of the gameplay filter uses a full-order dynamical model of the robot, both at synthesis and at deployment, which enables it to enforce safety across a broad range of motions and operating conditions.

## 2 Preliminaries: Robust Robot Safety in an Operational Design Domain

We wish to ensure the safe operation of a robot with potentially high-order nonlinear dynamics under a wide range of environments and task specifications, which may be unknown at design time. Formally, we consider a robotic system with uncertain discrete-time dynamics

$$x_{k+1} = f(x_k, u_k, d_k), \tag{1}$$

where, at each time step $k \in \mathbb{N}$, $x_k \in \mathcal{X} \subseteq \mathbb{R}^{n_x}$ is the state of the system, $u_k \in \mathcal{U} \subset \mathbb{R}^{n_u}$ is the bounded control input (typically from a control policy $\pi^u \in \Pi^u : \mathcal{X} \to \mathcal{U}$), and $d_k \in \mathcal{D} \subset \mathbb{R}^{n_d}$ is a disturbance input, unknown *a priori* but bounded by a compact $\mathcal{D}$. While the control bound $\mathcal{U}$ encodes actuator limits, the disturbance bound $\mathcal{D}$ is a key part of the *operational design domain* (ODD).

**Operational Design Domain.** The ODD can be viewed as social contract between the system operator and the public, delineating the set of conditions under which the robotic system is required to function correctly and safely [47]. In this paper, we are interested in *robust safety*, where the disturbance (or "domain") bound $\mathcal{D}$ may encode a range of potential perturbations like wind or contact forces, environmental parameters like terrain friction, manufacturing tolerances, variations in actuator performance and state estimation accuracy, and other factors contributing to designer uncertainty about future deployment conditions and modeling error. The ODD further specifies a deployment set $\mathcal{X}_0 \subset \mathcal{X}$ of allowable initial states (for example, the robot is always turned on while static on flat ground) and, crucially, a *failure set* $\mathcal{F} \subset \mathcal{X}$, which characterizes all configurations that the system state must never reach, such as falls or collisions. The required safety property can then be succinctly expressed as:

$$\forall x_0 \in \mathcal{X}_0, \forall k \geq 0, \forall d_0, \ldots, d_k \in \mathcal{D}, \quad x_k \notin \mathcal{F}, \tag{2}$$

that is, once deployed in an admissible initial state, the robot must stay clear of the failure set for any realization of the domain uncertainty.

**Safety Filter.** Explicitly ensuring the safety property in the synthesis of every robot task policy $\pi^{\boxtimes}$ can be impractically cumbersome, especially for increasingly general-purpose robotic systems with broad ODDs. Instead, we aim to relieve task policies of the burden of safety by augmenting them with a safety filter $\phi$ that depends on the robot's ODD but *not* on the task specification. Rather than directly applying the proposed task action $u_k = \pi^{\boxtimes}(x_k)$ from each state $x_k$, the robot executes[1]

$$u_k = \phi(x_k, \pi^{\boxtimes}) . \tag{3}$$

The safety filter's role is to prevent the execution of any candidate actions that would jeopardize future safety, while also avoiding spurious interventions that unnecessarily disrupt task progress. In

---

[1] For the scope of this paper, we assume that the robot maintains an appropriately accurate estimate of its dynamical state through onboard perception. We make two observations: First, moderate state estimation errors typical in many robotic systems can be absorbed by inflating the failure set $\mathcal{F}$ and dynamical uncertainty $\mathcal{D}$. Second, more substantial state uncertainty, e.g., induced by sensor faults, occluding objects, or multiagent interaction, may be handled with information-space safety filters, a subject of ongoing research [48–50].

fact, for any well-defined ODD there exists a *perfect safety filter* that allows every safe candidate action and overrides every unsafe one, robustly enforcing (2) with no overstepping [13, Prop. 1]. Formally, a perfect safety filter only disallows actions that may cause the state to exit the *maximal safe set* $\Omega^* \subset \mathcal{X}$, the set of all states from which there *exists* a control policy that can enforce (2). While computing such a perfect filter is known to be intractable for most practical systems [7], we aim to synthesize effective safety filters that allow robots significant freedom to perform a wide range of tasks (including online learning and exploration) while maintaining safety across their ODD. Intuitively, we would like to obtain a safety filter that robustly keeps the robot inside a conservative safe set $\Omega \subseteq \Omega^*$ as close as possible to the theoretical $\Omega^*$. Our proposed method uses game-theoretic reinforcement learning and faster-than-real-time gameplay simulation to *approximate* a perfect safety filter for any given robot ODD, targeting the robot's full dynamic envelope, in contrast with existing reduced-order filters, which aim to enforce safety within a significantly smaller set $\Omega$.

**Reach–Avoid Safety Game.** Whether it is possible for the robot to robustly maintain safety, as in (2), can be seen as the categorical (true/false) outcome of a *game of kind* between the robot's controller and an adversarial disturbance that aims to drive it into the failure set. In turn, this result can be encoded implicitly through a *game of degree* with a continuous outcome (for example, the closest distance that will separate the robot and any obstacle). In particular, for the purposes of predictive safety filtering, we consider a sufficient finite-time condition for all-time safety: it is enough for the robot to reach a known controlled-invariant set $\mathcal{T} \subset \mathcal{F}^c$ (for example, coming to a stable stance) in $H$ steps without previously entering the failure set $\mathcal{F}$. Once there, the robot can switch to a policy $\pi^{\mathcal{T}}$ that keeps it in $\mathcal{T}$ indefinitely. This induces a reach–avoid game [17, 32] with outcome

$$J_k^{\pi^u, \pi^d}(x) := \max_{\tau \in [k, H]} \min \left\{ \ell(x_\tau), \min_{s \in [k, \tau]} g(x_s) \right\} \tag{4}$$

where $g$ and $\ell$ are the (Lipschitz) failure and target margins, satisfying $g(x) < 0 \Leftrightarrow x \in \mathcal{F}$, $\ell(x) \geq 0 \Leftrightarrow x \in \mathcal{T}$. The outcome summarizes the aforementioned condition for all-time safety: For any given $\tau \in [0, H]$, if we previously enter the failure set $\mathcal{F}$, $g(x_\tau) < 0$, then for $k \in [0, \tau]$, $J_k^{\pi^u, \pi^d}(x) < 0$, denoting that past failure overrides future successes. The value function of this game satisfies the reach–avoid Isaacs equation

$$V_k(x) = \max_u \min_d \min \left\{ g(x), \max \left\{ \ell(x), V_{k+1}(f(x, u, d)) \right\} \right\}, \tag{5a}$$

$$V_H(x) = \min \left\{ \ell(x), g(x) \right\}, \tag{5b}$$

and the robot's controller is guaranteed a winning strategy from any state where $V_0(x) \geq 0$}.

# 3 Predictive Gameplay Safety Filters

## 3.1 Offline Gameplay Learning

We extend the Iterative Soft Adversarial Actor–Critic for Safety (ISAACS) scheme [38] to reach–avoid games (4), approximately solving the infinite-horizon counterpart of the Isaacs equation (5).

**Simulated Adversarial Safety Games.** At every time step of gameplay, we record the transition $(x, u, d, x', \ell', g')$ in the replay buffer $\mathcal{B}$, with $x' := f(x, u, d)$, $\ell' := \ell(x')$ and $g' := g(x')$.

**Policy and Critic Networks Update** The core of the proposed offline gameplay learning is to find an approximate solution to the time-discounted infinite-horizon version of (5). We employ the Soft Actor–Critic (SAC) [51] framework to update the critic and actor networks with the following loss functions. We update the critic to reduce the deviation from the Isaacs target[2]

$$L(\omega) := \mathbb{E}_{(x, u, d, x', \ell', g') \sim \mathcal{B}} \left[ (Q_\omega(x, u, d) - y)^2 \right],$$

$$y = \gamma \min \left\{ g', \max \left\{ \ell', Q'_\omega(x', u', d') \right\} \right\} + (1 - \gamma) \min \left\{ \ell', g' \right\} \tag{6a}$$

---

[2]Deep reinforcement learning typically involves training an auxiliary target critic $Q'_\omega$, with parameters $\omega'$ that undergo slow adjustments to align with the critic parameters $\omega$. This process aims to stabilize the regression by maintaining a fixed target within a relatively short timeframe.

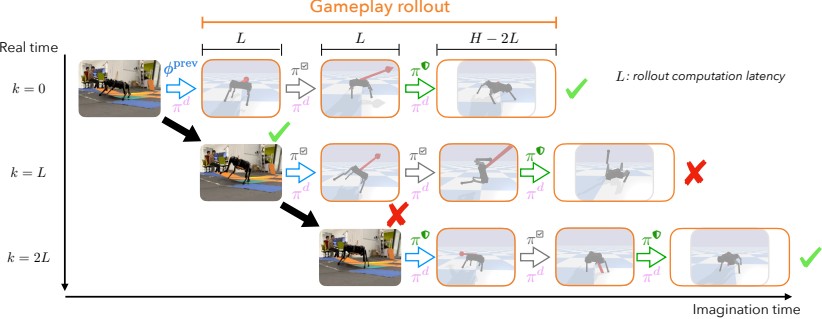

**Figure 2:** The $L$-step gameplay filter evaluates $\pi^{\boxtimes}$ for $L$ steps. The rollout started at $k\!=\!0$ returns a win result by $k\!=\!L$, so the safety filter allows $\pi^{\boxtimes}$ to proceed for $k\!=\!L,\dots,2L-1$. Meanwhile, the next rollout, initiated at $k=L$, predicts a defeat, so the filter selects $\pi^{\mathbf{\Phi}}$ from $2L$ to $3L-1$.

with $u' \sim \pi_\theta(\cdot \mid x')$, $d' \sim \pi_\psi(\cdot \mid x')$. We update control and disturbance policies following the policy gradient induced by the critic with entropy regularization:

$$L(\theta) := \mathop{\mathbb{E}}_{(x,d)\sim\mathcal{B}} \Big[ - Q_\omega(x,\tilde{u},d) + \alpha^u \log \pi_\theta(\tilde{u} \mid x) \Big], \tag{6b}$$

$$L(\psi) := \mathop{\mathbb{E}}_{(x,u)\sim\mathcal{B}} \Big[ Q_\omega(x,u,\tilde{d}) + \alpha^d \log \pi_\psi(\tilde{d} \mid x) \Big], \tag{6c}$$

where $\tilde{u} \sim \pi_\theta(\cdot \mid x)$, $\tilde{d} \sim \pi_\psi(\cdot \mid x)$, and $\alpha^u, \alpha^d$ are hyperparameters incentivizing exploration (entropy in the stochastic policies), which decay gradually in magnitude through the training.

Following the ISAACS scheme, we jointly train the safety critic, controller actor and disturbance actor through (6). For better learning stability, the controller actor can be updated at a slower rate (only every $\tau \geq 1$ disturbance updates), consistent with the asymmetric information structure of the game, and a leaderboard of best-performing controllers and disturbances can be maintained to mitigate mutual overfitting to the latest adversary iteration [38].

## 3.2 Online Gameplay Filter

This section demonstrates how the synthesized reach–avoid control actor $\pi_\theta$ and disturbance actor $\pi_\psi$ can be systematically used at runtime to construct highly effective safety filters for general nonlinear, high-dimensional dynamic systems. The gameplay rollout considers applying the candidate task policy $\pi^{\boxtimes}$ followed by the learned fallback policy $\pi^{\mathbf{\Phi}}$, with the whole rollout under attack by the learned disturbance policy $\pi_\psi$ to check if accepting the candidate action from task policy $\pi^{\boxtimes}$ will lead to an inevitable failure even if we then apply our best-effort attempt to maintain safety. The gameplay outcome follows the reach–avoid outcome defined in (4). A runtime gameplay filter can then be defined with the simple switching rule:

$$\phi(x,\pi^{\boxtimes}) = \begin{cases} \pi^{\boxtimes}, & \Delta^{\mathbf{\Phi}}(x,\pi^{\boxtimes}) = 1, \\ \pi^{\mathbf{\Phi}}, & \Delta^{\mathbf{\Phi}}(x,\pi^{\boxtimes}) = 0, \end{cases} \quad \Delta^{\mathbf{\Phi}}(x,\pi^{\boxtimes}) := \mathbb{1}\Big\{ \begin{array}{l} \exists \tau \in \{1,\dots,H\},\ \hat{x}_\tau \in \mathcal{T} \wedge \\ \forall s \in \{1,\dots,\tau\},\ \hat{x}_s \notin \mathcal{F} \end{array} \Big\} \tag{7a}$$

with $\hat{x}_0 = x$, $\hat{x}_{\tau+1} = f(\hat{x}_\tau, \hat{u}_\tau, \pi_\psi(\hat{x}_\tau)), \tau \geq 0$, and

$$\hat{u}_\tau = \begin{cases} \pi^{\boxtimes}(\hat{x}_\tau), & \tau = 0, \\ \pi^{\mathbf{\Phi}}(\hat{x}_\tau), & \tau \in \{1,\dots,H-1\}, \end{cases} \qquad \pi^{\mathbf{\Phi}}(x) = \begin{cases} \pi_\theta(x), & x \notin \mathcal{T}, \\ \pi^{\mathcal{T}}(x), & x \in \mathcal{T}. \end{cases} \tag{7b}$$

That is, if the gameplay monitor predicts a *win* (the simulated trajectory safely reaches the target set), the filter selects the task policy $\pi^{\boxtimes}$; otherwise, the filter selects the fallback safety policy $\pi^{\mathbf{\Phi}}$.

In practice, the computation of a full gameplay rollout may span multiple time steps (i.e., multiple control policy executions). In that case, the filter in (7) can be extended to a multi-step variant in which decisions are made by the filter every $L$ steps, appropriately accounting for the rollout computation latency. Fig. 2 illustrates the gameplay safety filter logic with $L$-step latency.

# 4 Experimental Evaluation

We run hardware experiments and an extensive simulation study, focusing on quadruped robots as an informative platform but stressing that our proposed methodology is general and can be applied to other types of robots. We aim to evaluate the extent to which the synthesized gameplay filters can maintain safety within the ODD specified at training, generalize beyond the ODD, and avoid unnecessarily impeding task execution. Specifically, we use a 50 N force applied to the robot's torso in training, with failure defined as any time a non-foot part contacts the ground. We test the robots under two conditions: tugging forces on flat terrain (similar to ODD) and irregular terrain (out-of-ODD). The task of moving from a start location to a goal across a terrain is evaluated on the Ghost Robotics Spirit S40 and Unitree Go2 (Fig. 1). We also conduct ablation studies to investigate the importance of reach–avoid reinforcement learning (RARL) and adversarial self-play in the filter synthesis and of the gameplay rollout in the filter's runtime monitoring by considering three prior reinforcement learning algorithms: (1) standard SAC [51] with (sparse) reward defined as $+1$ inside $\mathcal{T}$, $-1$ inside $\mathcal{F}$, and 0 everywhere else; (2) single-agent RARL [32] with and without domain randomization (DR); and (3) adversarial SAC with the above sparse reward. We also compare to a critic (value-based) filter, which intervenes when $Q_\omega(x, \pi^{\boxtimes}) < \epsilon$ with the threshold $\epsilon$ determined by running a parameter sweep and used in all experiments. Implementation details are in Appendix B.

## 4.1 Physical Results

**Safe walking within and beyond the ODD.** We evaluate the effectiveness of our proposed gameplay filter in terms of both safety and disruption of task performance. We run similar experiments with baseline methods for rough comparison purposes but caution that, due to the impossibility of reproducing identical conditions, these results should not be taken as a fine-grained quantitative comparison between methods. Such a comparison is conducted at scale, albeit in simulation, in Section 4.2.Table 1 shows the results for the S40 robot, subject to tugging forces and irregular terrain (not considered in the ODD), and Table 2 shows the results for the Go2 robot under a larger range of tugging forces (up to $4\times$ the ODD bound). Our proposed gameplay safety filter is remark-

Table 1: We evaluate S40 walking under tugging forces (modeled) and irregular terrain (unmodeled). We record the fraction of safe (fall-free) runs, the frequency of filter interventions (as a fraction of all time steps) and the average time taken to reach the goal. In the tug test, we also report the fraction of withstood attacks (tugs separated by at least 1.0 s), both overall and restricted to those roughly (within 10% error) inside the ODD bound of 50 N, and the peak force statistics in successful/failed runs. The gameplay filter completes all terrain runs safely and withstands almost all tugs (even out-of-ODD) without impractically hindering task progress.

| Policy | Tugging Forces | | | | | | | | Irregular Terrain | | |
| --- | --- | --- | --- | --- | --- | --- | --- | --- | --- | --- | --- |
| | | | Successful Runs | | | | Failed Runs | | | Successful Runs | |
| | Safe/All Runs | Withstood Attacks All (within 110% ODD) | Filter Freq. | $T_{\text{goal}}$ | $F_{\text{avg}}^{\text{peak}}$ | $F_{\text{max}}^{\text{peak}}$ | $F_{\text{avg}}^{\text{peak}}$ | $F_{\text{min}}^{\text{peak}}$ | Safe/All Runs | Filter Freq. | $T_{\text{goal}}$ |
| $\phi^{\text{game}}$ | 7/10 | 53/56 (33/35) | 0.17 | 26.3 | 67.5N | 70.5N | 59.8N | 52.7N | 10/10 | 0.19 | 41.2 |
| $\phi^{\text{critic}}$ | 4/10 | 22/28 (10/15) | 0.10 | 26.8 | 73.7N | 80.9N | 53.6N | 40.0N | 5/10 | 0.22 | 33.5 |
| $\pi^{\boxtimes}$ | 0/10 | 6/16 (1/5) | – | – | – | – | 56.5N | 41.4N | 5/10 | – | 16.4 |

Table 2: We evaluate Go2 under tugging forces, comparing our gameplay filter against the robot's production-grade walking policy. The gameplay filter withstood stronger tugs and had fewer failures. It remained safe for all 5 runs in which tugs remained within the ODD bound (50 N) and only failed under forces of at least $2\times$ the ODD bound (while resisting *some* forces of over $4\times$ the ODD bound)

| Policy | Tugging Forces | | | | |
| --- | --- | --- | --- | --- | --- |
| | | Successful Runs | | Failed Runs | |
| | Safe/Total Runs All Runs (Runs within ODD) | $F_{\text{avg}}^{\text{peak}}$ | $F_{\text{max}}^{\text{peak}}$ | $F_{\text{avg}}^{\text{peak}}$ | $F_{\text{min}}^{\text{peak}}$ |
| $\phi^{\text{game}}$ | 8/10 (5/5) | 42.4N | 215N | 105.7N | 104.4N |
| $\pi^{\boxtimes}$ | 0/10 (0/10) | – | – | 32.7N | 15.3 N |
| $\pi^{\boxtimes}_{\text{built-in}}$ | 7/10 (5/5) | 23.9N | 134.7N | 106.1N | 94.3N |

Table 3: Maximum force magnitude withstood by the S40 with various policies[†] at different tugging angles. Our learned fallback $\pi^{\mathbb{O}}$ outperforms the task policy and other safety fallback baselines and has comparable robustness to the policy used in the target set.

| Algorithm | Maximum Force | | | |
| --- | --- | --- | --- | --- |
| | Left | | Right | |
| | Low | High | Low | High |
| $\pi^{\mathbb{O}}$ | 87.1N* | 61.1N* | 99.3N* | 59.1N* |
| $\pi_\theta$ | 100.5N* | 150.3N* | 121.6N* | 121.9N* |
| RARL + DR | 46.4N | 43N | 57.2N | 72.1N* |
| $\pi^{\boxtimes}$ | 83.2N | 96.9N | 82.8N* | 59N |
| $\pi^{\mathcal{T}}$ | 151.9N* | 173.7N* | 140.3N* | 142.6N* |

[†] Safety policies from reward-based RL and ISAACS with the avoid-only objective fail immediately before applying force.
[*] The policy was able to withstand this magnitude of force. Because the policy made the quadruped move in the tugging direction, we were not able to apply a larger force in 10 pull attempts.

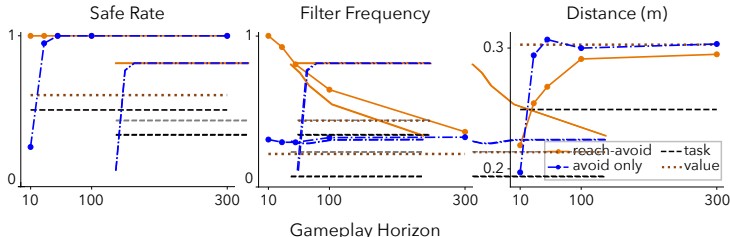

Figure 3: Sensitivity analysis of gameplay horizon and monitoring criterion. The proposed gameplay safety filter, utilizing the reach–avoid criterion, maintains a 100% safe rate (fraction of failure-free runs) for all gameplay horizons. As the horizon shortens, the gameplay filter with reach–avoid criterion only experiences a decrease in filter efficiency without compromising safety. In contrast, the filter with avoid-only criterion shows more safety violations than the task policy under short horizons. Overall, our proposed gameplay filter shows both higher safe rates and distance traveled (measured horizontally between the initial and final position) than the critic safety filter and task policy.

ably robust across robot platforms and test conditions; while not unbeatable outside of the specified ODD, it still withstands large tugging forces before violating the safety constraints. Importantly, the gameplay filter does not disproportionately interfere with the task-oriented actions: it maintains comparable filter frequency and task performance as the critic filter while drastically reducing safety failures. Fig. 1 shows the gameplay filter in action on the S40, dynamically counterbalancing tugs or springing into a wide stance. Time plots of tugging forces in all S40 runs are given in Appendix D.

**External forces.** We measure the maximum tugging force withstood by various safety policies and filters, reported in Table 3. We pull the quadruped from different directions, with "low" indicating angles in the range $[-0.1, 0.4]$ rad, and "high" in $[0.5, 1.0]$ rad. The employed $\pi_\theta$ can withstand 150 N from all directions, but the non-game-theoretic counterpart (RARL+DR) is vulnerable to the tugging from the left and can only withstand 43 N. This suggests that DR struggles to capture the worst-case realization of disturbances in a bounded class. This arises from its inherent nature: as the dimension of the disturbance input increases, the likelihood of the random policy simulating the worst-case disturbance decreases exponentially. Further, we notice the reward-based RL baselines and ISAACS with avoid-only objective fail almost immediately by overreacting and flipping over. Reach–avoid policies behave more robustly by bringing the robot to a stable stance. We also include tests for task policy $\pi^{\boxtimes}$ and the fixed-pose policy $\pi^{\mathcal{T}}$ (used when the state is in the target set). We observe that ISAACS control actor is strictly better than $\pi^{\boxtimes}$ and is comparable to $\pi^{\mathcal{T}}$.

## 4.2 Simulated Results

**Bespoke ultimate stress test (BUST).** To test each policy's robustness when taken to the limit, we RL-train a *specialized* adversarial disturbance $\pi^*_\psi$ to exploit its safety vulnerabilities (Table 4).

For each robot–disturbance policy pair, we play 1,000 finite horizon games and record the safe rate—overall fraction of failure-free runs. All pairs use the same set of 1,000 initial states. We observe that $\pi^{\boxtimes}$ is vulnerable to all $\pi^*_\psi$, while the proposed gameplay filter is only exploited by its associated

Table 4: We perform a bespoke ultimate stress test (BUST) of each control scheme in simulation by explicitly RL-training a *specialized* adversarial disturbance to find and exploit its vulnerabilities. We also sample random disturbances uniformly from $\mathcal{D}$ ($\pi^{\text{rnd}}$) or from its extreme points ($\pi^{\text{rnd},+}$), and observe that the domain randomization test is far less challenging (all safety methods excel).

| | $\pi_\psi^*(\pi_\theta)$ | $\pi_\psi^*\left(\pi^{☑}\right)$ | $\pi_\psi^*(\phi^{\text{game}})$ | $\pi_\psi^*\left(\phi^{\text{critic}}\right)$ | $\pi^{\text{rnd}}$ | $\pi^{\text{rnd},+}$ |
|---|---|---|---|---|---|---|
| $\pi_\theta$ | 0.37 | 0.38 | 0.17 | 0.44 | 0.88 | 0.85 |
| $\pi^{☑}$ | 0.0 | 0.0 | 0.0 | 0.0 | 0.03 | 0.03 |
| $\phi^{\text{game}}$ | 0.42 | 0.35 | 0.03 | 0.45 | 0.84 | 0.89 |
| $\phi^{\text{critic}}$ | 0.37 | 0.34 | 0.10 | 0.44 | 0.86 | 0.86 |

BUST disturbance $\pi_\psi^*(\phi^{\text{game}})$. Further, the robustness of $\phi^{\text{game}}$ pushes $\pi_\psi^*(\phi^{\text{game}})$ to learn effective attacks that also exploit other policies (the third column has the lowest safe rates compared to other columns across the board). The last two columns show the safe rate under random disturbances. All safety filters and safety policies remain at remarkably high safe rates, suggesting that our adversarial BUST evaluation method establishes a more demanding safety benchmark for policies than DR.

**Sensitivity analysis: reach–avoid criteria vs. avoid-only.** We evaluate the significance of using reach–avoid criteria in the gameplay filter by performing a sensitivity analysis of the horizon in the imagined gameplay. Fig. 3 shows that the gameplay filter with reach–avoid criteria still remains 100% safe rate even when the gameplay horizon is short ($H = 10$). In contrast, an "avoid-only" gameplay filter that only requires not reaching $\mathcal{F}$ for $H$ steps incurs more safety violations as the horizon decreases. The difference is due to shorter imagined gameplay resulting in more frequent filter intervention for reach–avoid criteria but overly optimistic monitoring for avoid-only criteria (oblivious to imminent failures beyond $H$). Further, as the gameplay horizon increases, the reach–avoid gameplay filter's intervention frequency decreases.

# 5 Conclusion

This work presents a game-theoretic learning approach to synthesize safety filters for high-order, nonlinear dynamics. The proposed gameplay safety filter monitors system safety through imagined games between its best-effort safety fallback policy and a learned virtual adversary, aiming to realize the worst-case uncertainty in the system. We validate our approach on two different quadruped robots under strong tugging forces and unmodeled irregular terrain while maintaining zero-shot safety. An exhaustive simulation study is also performed to rigorously stress-test the approach and quantify its reliability and conservativeness.

**Limitations.** Despite the strong empirical robustness in both simulated and physical experiments, we do not have strong theoretical guarantees on convergence of offline gameplay learning, and therefore learned disturbance policy can in general be expected to behave suboptimally in at least certain regions of the state space. Naturally, the potential implications are quite serious, since a suboptimal (not-truly-worst-case) disturbance model may lead the gameplay rollout to erroneously conclude that a proposed course of action is safe, only to then be met by an ODD realization that unexpectedly drives the robot into a catastrophic failure state. Without strong theoretical assurances that for now remain elusive, this is not a method that should be placed in sole charge of a truly safety-critical system where an eventual catastrophic failure can carry inadmissible cost.

The remarkably high effectiveness demonstrated by the gameplay filter across various within-ODD experiments and even under out-of-ODD conditions could indicate that this new type of filter does in fact enjoy desirable properties yet to be established. This calls for future theoretical work at the intersection of game-theoretic reinforcement learning and nonlinear systems theory. In parallel, we see an opportunity for application-driven research to leverage the computational scalability and *de facto* robustness of gameplay filters to tackle ongoing challenges in robot learning, for example for safe acquisition of novel skills as well as rapid detection of shifts in operating conditions enabling safe runtime adaptation of ODD assumptions.

**Acknowledgments**

This work has been supported in part by the Google Research Scholar Award and the DARPA LINC program. The authors thank Zixu Zhang for his help in preparing the Go2 robot for experiments.

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

# Appendices

## A  Frequently Asked Questions

We discuss some design choices and implications of our method using an informal FAQ format.

**Why choose worst-case safety and not probabilistic analysis?**  Although not as established in the robot learning community, robust/worst-case formulations are widely used across engineering. Their key advantage is that they can enforce systematic handling of *all* scenarios in a well-defined class, even if some of them are highly unlikely—e.g., the robot must withstand all (rather than most) external forces of up to 50 N, even the unlucky push that happens to maximally disturb its stance. This is consistent with much of the safety analysis found in bridges, elevators, automobiles, aircraft, and other safety-critical engineering systems, in great part because it facilitates a clear-cut social contract between their designers and the broader public. For example, we do not certify elevators for 95% of loads up to 300 kg or bridges for 99% of earthquakes up to magnitude 8, but rather all such loads and earthquakes, and we treat any loss of safety within the specified bounds as a serious failure to comply with the promise made to society. As robots and autonomous systems become more widely deployed, we argue that their safety should be certified and held to similar standards, at least in truly safety-critical settings where people could otherwise get hurt.

**Isn't worst-case safety too conservative to be useful?**  Actually, this is a common misconception. Robust/worst-case assessments are not intrinsically more or less conservative than probabilistic ones: this depends entirely on what set and distribution we choose to run these assessments against. The term "worst-case" doesn't mean a system must preserve safety in the worst *conceivable* scenario (whatever that means), but rather under all conditions—including the worst one—in a *specified* set. Worst-case safety lets designers and regulators draw this line (the ODD boundary), and it ensures that the system then maintains safety across all certified (in-ODD) conditions. If your robot's behavior is "too conservative" this means it's guarding against eventualities you don't really care about: just exclude them from your ODD. But, if you *do* want safety under these conditions, then your robot is not actually too conservative: it's doing what it should. ***With the gameplay filter, you are never left wondering: each time it overrides the task policy, it logs the specific future it's preempting.*** Then, only one question remains: did you or did you not want your robot to avoid that hypothetical crash? Worst-case safety is extremely powerful, and it lets you control exactly what situations your robot is required to handle. You just need to be ready to answer to some hard *what-if* questions.

**What does it mean for the proposed gameplay filter to approximate a perfect filter?**  If we had the exact solution to the Isaacs reach-avoid equation (5), our gameplay rollouts would be necessary and sufficient for safely reaching $\mathcal{T}$ in $H$ (or fewer) steps. Since $\mathcal{T}$ is typically chosen to be a broad, naturally reachable class of robot states (e.g., coming to a stable stance for a walking robot or pulling over for an autonomous vehicle), safely reaching $\mathcal{T}$ within a long enough horizon $H$ is possible as long as remaining safe is possible in the first place. In other words, the sufficient reach-avoid condition becomes a tight approximation of the all-time safety condition. We can observe this phenomenon in Fig. 3, where the reach-avoid filter's overstepping vanishes with long $H$.

**Why is computing a gameplay rollout better than just querying the learned reach–avoid critic?**  In theory, the critic should make fairly accurate predictions of game outcomes after training. In practice, we have found that it's often unreliable and/or overly conservative. A key advantage of the gameplay rollout is that the uncertainty linked to the learning-based safety analysis is much more structured: the robot's future safety fallback is perfectly predicted (since it will be implemented as-is), and the dynamics can be reliably simulated given players' actions, so all uncertainty falls on the learned disturbance. One very useful implication of this structure is that, even if the disturbance is suboptimally adversarial, a predicted gameplay rollout ending in a safety failure constitutes a valid certificate (i.e., a *proof*) that there exists an ODD realization in which the robot will violate safety if the filter does not intervene immediately. That is, we know the gameplay safety monitor isn't falsely crying wolf—we can't prove anything like that about the black-box neural safety critic's predictions.

**Why is reach–avoid preferable if it's more conservative than avoid-only?** This is an important aspect of predictive safety filtering and relates to a deeper tenet in safety engineering philosophy: whatever the safety boundary is (i.e., a strategy that is "just safe enough"), it is preferable to approach it from the safe side than from the unsafe side. In practice, we don't know a priori how many prediction steps $H$ we need to avoid being blindsided by future failures just beyond the lookahead horizon. When in doubt, it's preferable to risk being overly conservative than to risk losing safety.

**Having a terminal state constraint is common in MPC, how is reach–avoid different?** The use of a terminal controlled-invariant set in MPC is well established and ensures recursive feasibility. Our choice of reach–avoid over an avoid–only safety condition is an instance of the same principle. An important difference is that the (also well-established) reach–avoid condition gives our filter extra flexibility by allowing the gameplay trajectory to reach the forever-safe set $\mathcal{T}$ at any time within the horizon. This reduces conservativeness and often lets us to terminate the gameplay rollout early.

**How do you determine $\mathcal{T}$?** In practice, a suitable $\mathcal{T}$ is obtained from domain knowledge, offline computation, pre-deployment learning, or some combination, often in the form of a stability basin (region of attraction) around a desirable class of equilibrium points sufficiently away from failure. For example, most robots can be robustly stabilized around static or steady cruising configurations by comparatively simple linear feedback controllers (e.g., most modern walking robots ship with built-in controllers that can stabilize them around a default stance). Larger all-time safe regions may be found by (robust) Lyapunov analysis or even optimized through control Lyapunov functions.

**What are the implications of the choice of $\mathcal{T}$?** Broadly speaking, the larger the $\mathcal{T}$ we can characterize offline, the easier the job of the gameplay filter at runtime, and, potentially, the fewer steps we'll need to reach it from more dynamic configurations. In fact, in the extreme case, we could be remarkably lucky and find $\mathcal{T} = \Omega^*$, in which case, the gameplay filter's job is made much easier, since all candidate actions that are safe will keep the state in $\mathcal{T}$, immediately terminating the rollout check. Conversely, all actions that leave $\mathcal{T}$ are unsafe and the gameplay rollout will not be able to return to $\mathcal{T}$. In order to avoid initializing the gameplay filter from a no-win scenario, designers should ensure that $\mathcal{T}$ contains the range of expected robot deployment conditions ($\mathcal{X}_0$) in the ODD.

**Why aren't you using onboard cameras or lidar?** Our empirical focus in this paper is on demonstrating automatically synthesized safety filters that account for the full-order (36–D) walking dynamics of quadruped robots. We think that the simplest and clearest demonstration of this concept is by having the filter only consider the robot's own state (proprioception) without accounting for the environment, obstacles, etc. (exoception). That said, incorporating information about the robot's surroundings can be extremely valuable—and often critical—to safety. We are very excited by the scalability and generality that new safety approaches like the one we present in this paper seem to enjoy, and we expect they will soon unlock full-order safety filters that incorporate rich exoceptive information in real time, whether straight from raw sensor data or through intermediate representations provided by the perception and localization stack.

## B Implementation Details

**Robot hardware.** Both the Ghost Robotics Spirit S40 and Unitree Go2 have built-in IMUs to obtain body angular velocities and linear acceleration, and internal motor encoders to measure joint positions and velocities. The S40 has no foot contact sensing; the Go2 receives a Boolean contact signal for each foot. Neither robot's safety filter is given access to visual perception.

**Gameplay filter runtime implementation.** To easily deploy our gameplay safety filter across two different robots for the physical experiments, we encapsulate its computation inside a ROS service, which we run on an offboard computer. Each robot's onboard process calls this service wirelessly (approximately 3.5 times per second) passing its current state estimate and proposed course of action /newcandidate task policy; the offboard server then simulates a $H$-step gameplay for a fixed horizon (for us, $H = 300$ or 3 s) and returns a Boolean indicating which policy to use for the next $L$ time steps; our choice of $L = 10$ accounts for the wireless round trip, which makes up a significant fraction (approximately $70\%$) of the total latency.

We note that the computational resources used for the offboard computation are comparable to those available on current mobile robot platforms. In particular, the entire simulation and filter logic was run on one single core of an Intel i7-1185G7 processor at 3GHz. For comparison, the Go2 is equipped with a second computer (not used in our experiments) with a 6-core 8GB NVIDIA Jetson Orin Nano processor at 1.5 GHz. We estimate that the total latency of the gameplay filter run fully onboard the Go2 with the same simulator would be roughly comparable, possibly lower given the absence of a wireless roundtrip.

**Operational design domain.** The safety filter is computed for a fairly simple ODD, defined by the nominal robot simulator perturbed by forces of up to 50 N applied anywhere on the robot's torso; the disturbance adversary acts by a vector $d \in \mathcal{D} \subset \mathbb{R}^6$ encoding what force to apply and where. We intentionally limit the ODD to only consider flat ground. The failure set $\mathcal{F}$ is defined as all *fall* states, in which any non-foot robot part makes contact with the ground. The deployment set and controlled-invariant set $\mathcal{X}_0 = \mathcal{T}$ are chosen empirically to contain all four-legged stances with a lowered torso, around which the robot is robustly stable with a simple leg position controller $\pi^{\mathcal{T}}$.

**Test conditions.** The irregular terrain is a 2 m × 4 m area with a 15-degree incline along one edge, and two memory foam mounds, 5 cm and 15 cm high, positioned 1.8 m from each other. Tugging forces are applied manually through a rope, attached to the robot's torso and to a motion-tracked dynamometer with rated capacity of 500N, 0.1 N resolution and sampling rate of 1000 Hz and set to provide audiovisual alerts at 80% and 100% of the ODD limit

**Policies.** The learned control and disturbance actors, as well as the safety critics, are independent of the robot's absolute position $p_x, p_y, p_z$ and heading angle $\theta_z$; of these, only distance to the ground has an effect on the dynamics, but since it is hard to observe without vision, we do not make it available. In the case of the Go2 quadruped (but not the S40), the policies additionally depend on the discrete contact state, encoded as a Boolean (true/false) indicator for each foot. In simulation, each neural network policy receives as input the ground-truth state of the robot in the simulator; in hardware experiments, they instead receive a state estimate computed by the robot's on-board perception stack. Each policy is implemented by a fully-connected feedforward neural network with 3 hidden layers of 256 neurons, and critics have 3 hidden layers with 128 neurons. We handcraft a task policy using an inverse kinematics gait planner for forward/sideways walking. We use a low-level PD position controller that outputs torques $\tau^i = K_p(\delta\theta_{\mathrm{j}}^i) - K_d \cdot \omega_{\mathrm{j}}^i$ to the robot motor controller with $K_p, K_d$ the proportional and derivative gains.

**State and action spaces.** For the scope of this paper, we aim to construct a *proprioceptive* safety filter that relies on onboard estimation of the robot's kinematic state but *no exoceptive* information

(from camera, lidar, etc.) about the surrounding environment.[3] We encode the quadrupedal robots' state and action vectors as follows:

$$x := \left[ p_x, p_y, p_z, v_x, v_y, v_z, \theta_x, \theta_y, \theta_z, \omega_x, \omega_y, \omega_z, \{\theta_J^i\}, \{\omega_J^i\} \right],$$
$$u := \left[ \{\delta\theta_J^i\} \right],$$

with $p_x, p_y, p_z$ the position of the body frame with respect to a fixed reference ("world") frame; $v_x, v_y, v_z$ the velocity of the robot's torso expressed in (forward–left–up) body coordinates; $\theta_x, \theta_y, \theta_z$ the roll, pitch, and yaw angles of the robot's body frame with respect to the world frame;[4] $\omega_x, \omega_y, \omega_z$ the body frame's axial rotational rates; and $\theta_J^i, \omega_J^i, \delta\theta_J^i$ the angle, angular velocity, and commanded angular increment of the robot's $i^{\text{th}}$ joint.

The above constitutes a full-order state representation of the robot's idealized Lagrangian mechanics. A total of 18 generalized coordinates encode the 6 degrees of freedom of the torso's rigid-body pose in addition to the configuration of 3 rotational joints (hip abduction, hip flexion, and knee flexion) for each of the 4 legs; the robot's rate of motion is expressed through 18 corresponding generalized velocities, for a total 36 continuous state variables. We discuss discrete contact variables below.

The robot's control authority is achieved by independently modulating the torque applied on each of its 12 rotational joints by an electric motor; in modern legged platforms, these motors typically have dedicated low-level controllers, so our control policy sends a tracking reference to each motor controller rather than directly commanding a torque.

Finally, the disturbance is modeled as an external force that can act on any point of the robot's torso and in any direction of Euclidean space, with a bounded modulus. The specified range of admissible disturbance forces is discussed below.

**Black-box simulator(s).** The dynamical model is implemented by the off-the-shelf PyBullet physics engine [52] using the standardized robot description files made available by the manufacturers of each platform. Our method treats the simulator as black-box environment for both training and runtime safety filtering, allowing the engine and/or robot model to be easily swapped out. The generality and modularity of our approach is perhaps best illustrated by the fact that we synthesized and deployed the safety filter for the Go2 robot using *identical hyperparameter values* as for the S40 robot. Our only modification, other than replacing the robot model in the physics engine, was to append 4 state components to each neural network's input space to account for foot contact information; we note that even this straightforward addition is entirely optional, since we could have alternatively constructed a safety filter that simply disregarded the extra sensor data.

**Safety specification.** We are interested in preventing *falls*, understood as any part of the robot other than its feet making contact with the ground. To encode the failure set of all such falls with a simple margin function, we define a small number of critical points $\mathbf{p_c}$, including the 8 corners of a (tight) 3-D bounding box around the robot's torso as well as its four knee joints. The failure margin is

$$g(x) = \min \left\{ \min_i \{z_{\text{corner}}^i\} - \bar{z}_{\text{corner},g}, \ \min_i \{z_{\text{knee}}^i\} - \bar{z}_{\text{knee}} \right\},$$

with $z_{\text{corner}}^i$ the vertical distance to the ground of the $i^{\text{th}}$ robot body corner point and $z_{\text{knee}}^i$ the vertical distance to the ground of the $i^{\text{th}}$ robot knee point. The target (all-time safe set) is defined as a narrow neighborhood of a static stance with all four feet on the ground and a sufficiently lowered torso,

---

[3]Ranged perception can improve the robustness of walking controllers by sensing terrain geometry and texture, and it is strictly needed for ODDs including unmapped or moving obstacles. Full-order legged robot safety filters combining proprioception and exoception have significant potential and are ripe for investigation.

[4]For the purposes of this demonstration, we find that an Euler angle representation of body attitude performs adequately and makes the failure set straightforward to encode. In general, a quaternion-based representation may be preferable, avoiding the risk of computational issues in the neighborhood of singularities (at $\theta_y = \pm\frac{\pi}{2}$).

chosen so that the robot is robustly stable with a simple stance controller. The target margin is

$$\ell(x) = \min \Big\{ \bar{\omega} - |\omega_x|, \ \bar{\omega} - |\omega_y|, \ \bar{\omega} - |\omega_z|,$$
$$\bar{v} - |v_x|, \ \bar{v} - |v_y|, \ \bar{v} - |v_z|,$$
$$\bar{z}_{\text{corner},\ell} - \max_i\{z^i_{\text{corner}}\}, \ \bar{z}_{\text{foot}} - \max_i\{z^i_{\text{foot}}\}\Big\},$$

with $z^i_{\text{foot}}$ the vertical elevation of the $i^{\text{th}}$ robot foot relative to the ground. The threshold values we used for our failure and target set specification are as follows.

$$\bar{z}_{\text{corner},g} = 0.1 \text{ m} \qquad\qquad \bar{z}_{\text{knee}} = 0.05 \text{ m}$$
$$\bar{z}_{\text{corner},\ell} = 0.4 \text{ m} \qquad\qquad \bar{z}_{\text{foot}} = 0.05 \text{ m}$$
$$\bar{\omega} = 10°/\text{s} \qquad\qquad \bar{v} = 0.2 \text{ m/s}$$

**Uncertainty specification.** To account for uncertainty in the deployment conditions as well as general modeling error (or sim-to-real gap), our operational design domain (ODD) includes an external force that may push or pull any point on the robot's torso in any direction with a maximum magnitude of 50 N:

$$d = \left[ F_x, F_y, F_z, p^F_x, p^F_y, p^F_z \right] , \tag{8}$$

where $F = [F_x, F_y, F_z]$ represents the force vector applied at position defined by $p^F_x, p^F_y, p^F_z$ in the body coordinates $p^F_x, p^F_y \in [-0.1, 0.1]$, $p^F_z \in [0, 0.05]$ m. The red arrows in the imagined gameplay of Fig. 2 show examples of learned adversarial disturbance.

# C  Extended Evaluation

To further demonstrate the strengths of our approach and shed light on its superior scalability to complex robot dynamics, we compare the gameplay performance of the self-play-trained controller and disturbance policies *as training proceeds*. The results in Figures 4 and 5 suggest that the dense temporal difference signal in reach–avoid games plays a determining role in enabling data-efficient learning, while previously proposed safety methods that use reward-based RL with a (sparse) failure indicator consistently require more training episodes before starting to learn meaningfully robust safe control strategies.

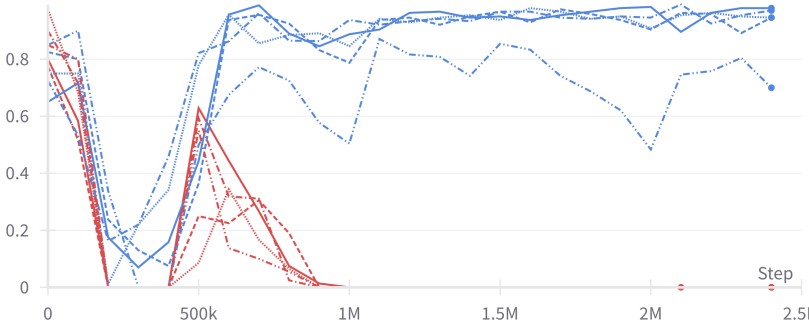

Figure 4: Safe rate achieved by the robot controller against the co-trained adversarial disturbance as the adversarial RL synthesis proceeds under reach-avoid objective (blue) and reward-based objective (red).

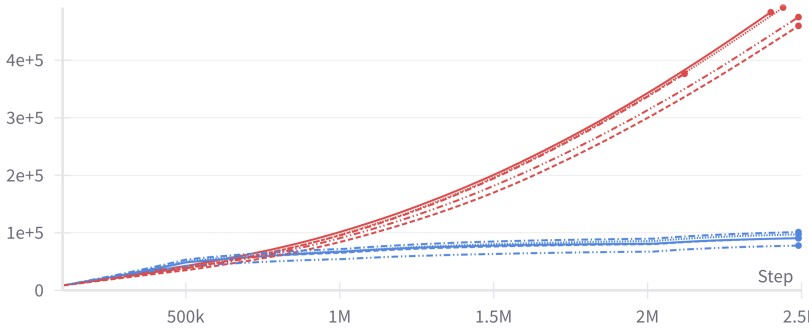

Figure 5: Cumulative safety violation count as the adversarial RL synthesis proceeds under reach-avoid objective (blue) and reward-based objective (red).

# D  Detailed Tugging Force Plots

We provide time plots for all runs of the tug test experiment on the S40 robot (summarized in Table 3), displaying the magnitude of the tugging force over the course of each trial. We present all 10 runs for each of the three evaluated control schemes: gameplay filter $\phi^{\text{game}}$, critic (value-based) filter $\phi^{\text{critic}}$, and unfiltered task policy $\pi^{\boxtimes}$. Each run is annotated to show individual attacks, defined as sequences of significant tug forces ($\geq 10\text{N}$) that are applied continually or close together in time (less than 1 s interruption within an attack). Conversely, distinct attacks are at least 1 s away from each other, to ensure that the effects of the previous attack have died off before the next one begins.

Looking at individual attacks within each run provides a more fine-grained insight on the performance of each control scheme under various disturbances (both within-ODD and out-of-ODD). Importantly, it allows us to attribute a safety failure to the attack that immediately preceded it in a given run, but mark all earlier attacks in the same run as safely handled.

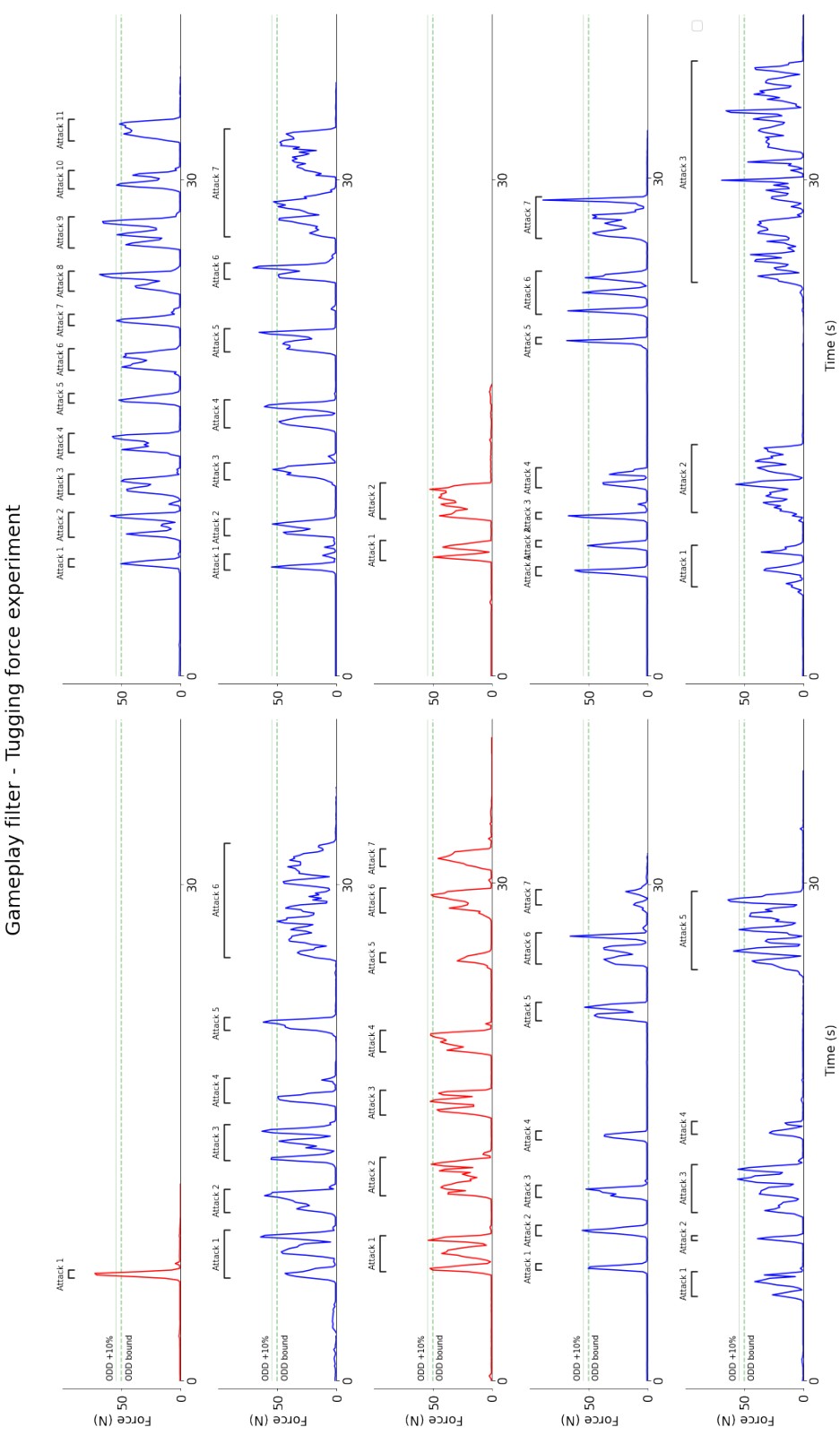

Figure 6: Time plots of the force applied in all 10 runs of the Spirit S40 physical tugging test using the gameplay filter. Across the 10 runs, there were 56 attacks, 35 of which were roughly (up to $10\%$ error) within the ODD bound of $50$ N. Plots in red indicate a failed run (ending in a fall), plots in blue indicate a safe (fall-free) run.

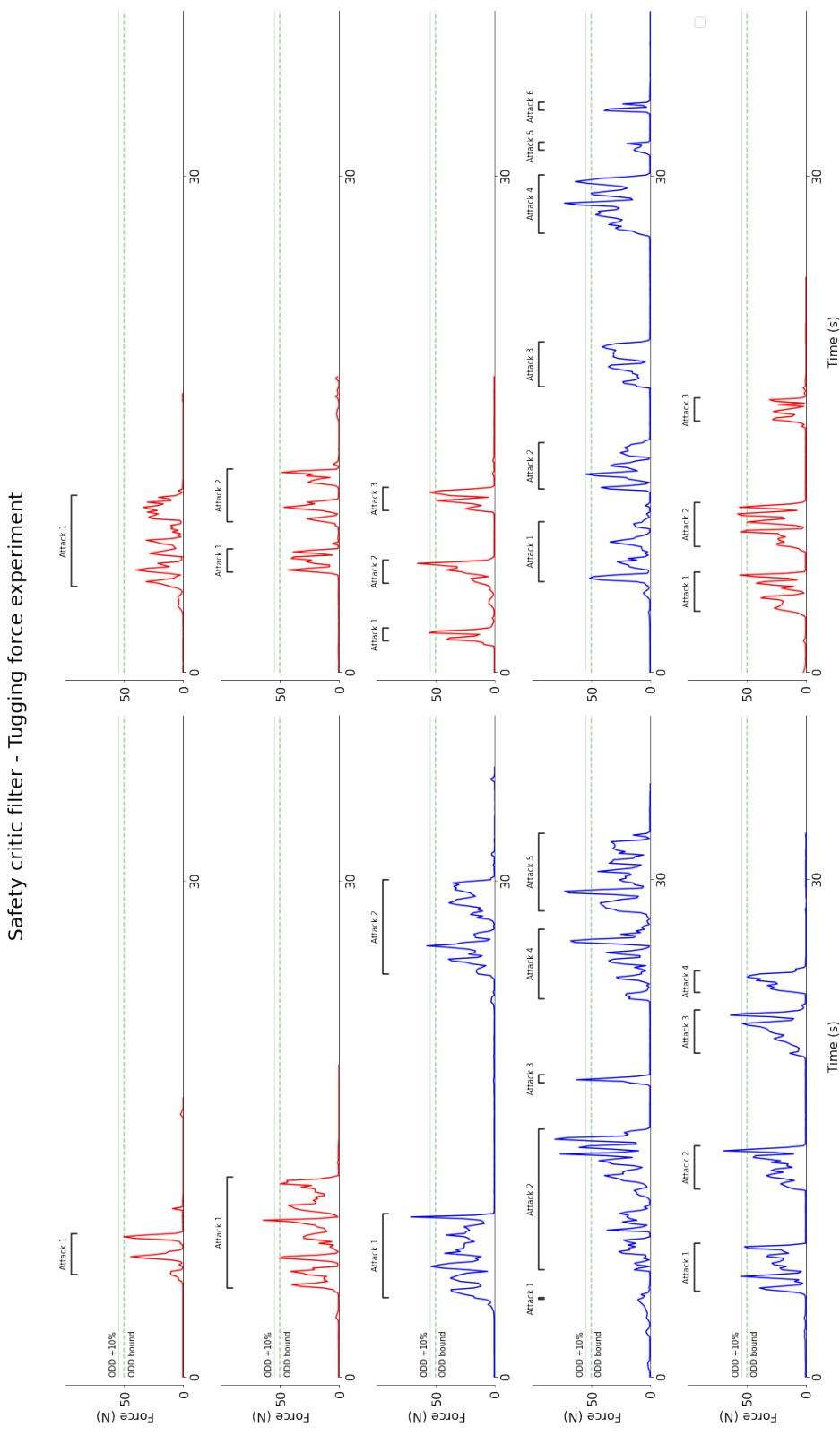

Figure 7: Time plots of the force applied in all 10 runs of the Spirit S40 physical tugging test using the critic (value-based) filter. Across the 10 runs, there were 28 attacks, 15 of which were roughly (up to 10% error) within the ODD bound of 50 N. Plots in red indicate a failed run (ending in a fall), plots in blue indicate a safe (fall-free) run.

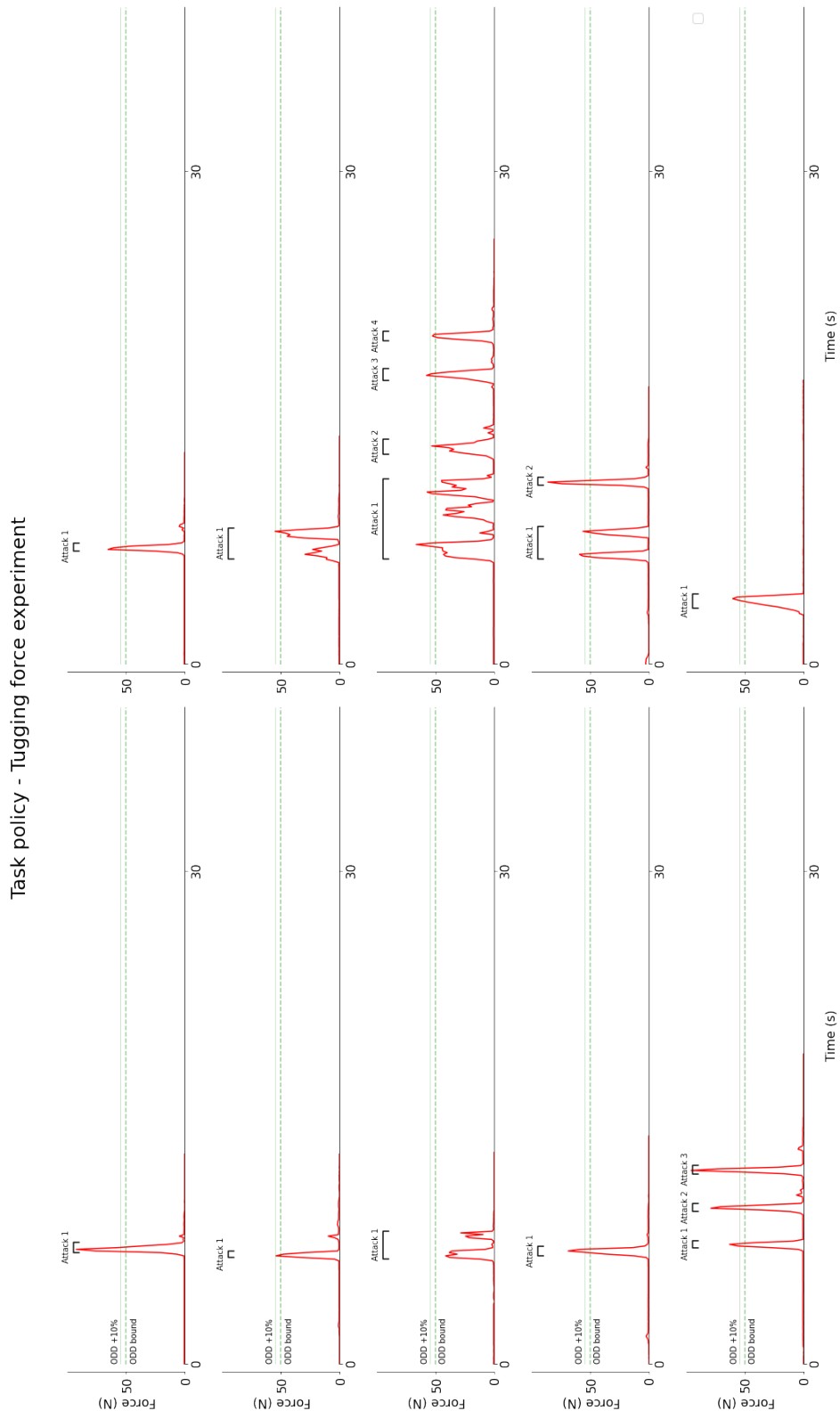

Figure 8: Time plots of the force applied in all 10 runs of the Spirit S40 physical tugging test using the gameplay filter. Across the 10 runs, there were 16 attacks, 5 of which were roughly (up to $10\%$ error) within the ODD bound of $50$ N. Plots in red indicate a failed run (ending in a fall), plots in blue indicate a safe (fall-free) run.

