# OpenReview forum: "Gameplay Filters: Robust Zero-Shot Safety through Adversarial Imagination"
_robot-learning.org/CoRL/2024/Conference — CoRL 2024_

### Official Review · Reviewer_tETa · 2024-07-18
**Review of Submission 725**

**Originality:** 4
**Technical Quality:** 4
**Clarity Of Presentation:** 5
**Potential Impact:** 3
**Recommendation:** 4
**Confidence:** 4

**Review:**

This paper presents an approach for getting a safety filter working on a high-dimensional physical robotic system through the use of adversarial RL based on HJ reachability.  The paper is generally well-written and contains both simulation and real-world experiments to prove the merits of their approach. The figures are also well-made and are helpful for better understanding the paper. The paper also contains a good overview of related approaches in safe control.

My main concern for this paper is the lack of clarity on the comparison of this approach to a value-based filter. In table 1, the authors show a comparison between $\phi^{game}$ (their proposed approach) and $\phi^{critic}$. There is a brief mention of the “critic filter” in line 198 but it is not clear whether this refers to one of the four baselines (specifically the two SAC baselines), or if this is totally separate. The most clear baseline for the proposed approach would be one that explicitly uses the value function trained alongside the safety and disturbance policies to determine when the fallback policy is used vs the task policy (i.e. if $V(x)<\epsilon$, use the fallback policy). If $\phi^{critic}$ is indeed this exact value-based safety filter then 1) this needs to be made more clear in the text and 2) the authors should spend some time explaining why their proposed approach works better than this style of safety filter—at the moment this is not mentioned in the text, and seems unintuitive why the gameplay filter would work significantly better than a value-based approach, since the value function is being trained alongside the safety and disturbance policies being used for imagined gameplay. If $\phi^{critic}$ is *not* this kind of value-based filter, the authors should also make this clear, and additionally compare their method to it.

The reach-avoid approach relies on having a known control-invariant set $\mathcal{T}$ and a known Lipschitz-continuous margin function $l(x)$ that tells you how far you are from it. This makes sense in the case of having one stable static configuration, but may be difficult to define more generally for a system where you would want to return to a dynamic trajectory (like having the robot running or a quadcopter flying forwards). If it’s not possible to hand-design a margin function for returning to an arbitrary dynamic motion, the proposed approach may be limited in its applicability to more dynamic robot environments.

The ODD (i.e. range of forces in Newtons applied to the system during training) is not explicitly listed in the experiments section. This makes it harder to understand the results in tables 1-3. In line 193, the authors note that the dynamometer has a rated capacity of 500N, but do not clarify if 500N is the maximum value of the ODD used for training.

Related to the previous point, the caption in Table 1 says “Note that not all tug runs were inside the ODD.” It would be nice to explicitly split up the results from this table based on whether the tug was inside or outside the ODD—this would help us understand how robust the proposed and baseline methods are to out-of-distribution disturbances.

The paper claims that the proposed gameplay filter is a “new class of predictive safety filter”, but it seems like this approach is essentially MPC applied to a safe control scenario, where instead of picking a control that minimizes a continuous future cost, there is just a binary cost for whether the future rollout results in failure. The inclusion of a trained adversary in this loop is certainly novel, but the authors may want to make less strong claims about the novelty of the full “gameplay filter” idea and talk a bit about its connection to MPC, particularly to literature on model predictive safety filters.

Small comments:

- On line 139, there is an extra }
- On line 182, it should say “both have” instead of “both has”

**Quality Of The Limitations Section:**

3

**Questions For Rebuttal:**

1. What is $\phi^{critic}$? This is not explicitly written out in the text, and it should be made clear if this is a value-based filter that uses the trained critic from the proposed adversarial RL framework to do the switching control. If it is such a filter, *why* does it perform worse than the proposed gameplay filter? If it is not such a filter, then the authors should do a comparison to a value-based filter using the trained value function.
2. How can you generally define a margin function $l(x)$ for a control-invariant set, which may not just be a single static pose? (e.g. walking or running gait for a quadruped or trim controls for an aircraft)
3. What is the ODD used for experiments, and can you split the results in table 1 for tugs in and out of the ODD?
4. There is no limitations section, please add one.

**Robotics Focus:**

4

**Summary Of Paper:**

This paper proposes a “gameplay filter” that consists of simulating a potential worst-case disturbance online and choosing actions from either a task-oriented policy or a fallback policy at each timestep depending on whether the simulated gameplay results in success or failure. The gameplay filter is trained using a game-theoretic adversarial RL approach that trains a reach-avoid safety policy and disturbance policy via self-play. The authors instantiate this problem for a quadruped robot trying to stay upright. When compared to other safety filters, their approach is more robust to disturbances both in simulation and in real-world experiments. This paper is ultimately one of the first approaches to successfully roll out a safety filter for a physical quadruped on a full-fidelity (36D) model of the robot, instead of relying on reduced-order dynamics models, which most prior work has done.

**Summary Of Recommendation:**

This paper constructs a novel safety filtering method based on HJ reachability that simulates potential worst-case disturbances online to stay robust to them. Their approach is shown to work well for a quadruped robot both in simulation and on two different hardware platforms.   Based on the revised manuscript and authors' comments in the rebuttals, my main concerns about the value-based filter and limitations section have been addressed, and my review has been adjusted accordingly. I continue to recommend accepting this paper.

---

### Official Review · Reviewer_ZswE · 2024-07-20
**Review of Submission 725**

**Originality:** 4
**Technical Quality:** 4
**Clarity Of Presentation:** 4
**Potential Impact:** 3
**Recommendation:** 3
**Confidence:** 4

**Review:**

## Strengths:
1. Novel use of game theoretic formulation in safety filters for robotics control.
2. Realtime safety filter with full order dynamics in forward simulation.

## Weaknesses:
1. Experiments can be more rigorous and better explained. For example, the disturbances/unmodeled terrain is not quantify to give readers an idea of what kind of disturbances are considered in the ODD.
2. SAC utilizes stochastic policies. The authors did not discuss the coverage of adversarial disturbance policies in the sense that what happens when the controller actor produces an out-of-distribution combination of state and action that the disturbance actor has not seen. This seems important since the claim is that a strategy that wins against the worst case wins against all others.
2. Clarity can be improved when explaining some key concepts.
3. The organization of the experimental section can be significantly improved for better understanding of the results.
4. There is no limitation section.

Please refer to the following additional comments for more details:

### Additional Comments:
1. It’ll be helpful to at least have some quantification of how irregular/unmodeled the terrain is. The only example is shown in Figure 1, which only shows a slight slant in the floor.
2. An intuitive explanation (like the one in [17]) for Equation (4) will be helpful for explaining the reach-avoid game value.
3. The “safe rate” in Table 4 is unspecified. Is it the percentage of timesteps, or the percentage of full rollouts?
4. The third figure in Figure 3 is not clearly referenced/explained in the text.
5. In the simulation experiment (BUST), what does “specialized” mean? This seems like an important detail that is omitted.
6. In the results, some baselines that were compared to were not explained. ($\pi^{\mathcal{T}}$). In addition, some explanation on why each baseline is chosen in comparison will help highlight the improvements.

**Quality Of The Limitations Section:**

3

**Questions For Rebuttal:**

1. How are tugging forces decided in each experiment? Are they the same/similar across different baseline comparisons? If not, are there any metrics that can show that the experiments are still valid?
2. Could you provide some quantification on how "unmodeled" the terrains are in the experiments?
3. How do you deal with the cases when the stochastic policies produce "false positive" winning cases against the disturbance policy during forward simulation?
4. What does "specialized" mean in the BUST experiment?
5. What is the "Distance" in the third figure of Figure 3?

**Robotics Focus:**

4

**Summary Of Paper:**

This paper introduces a real-time safety filter for legged robot locomotion via simulated adversarial self-play. A disturbance actor that simulates the sim-to-real gap is trained alongside and against a control actor using SAC-like scheme. Online, gameplay filters forward simulates the robot against learned disturbance actors, and uses the learned control policy as the fallback policy. Then a simple switching rule is used to select control actions.

**Summary Of Recommendation:**

Weak Accept, organization and clarity of the experiment section can be significantly improved, limitation section is missing and needs to be added. After rebuttal, clarity is improved and the limitation section addresses my biggest concern, and ratings are adjusted accordingly.

---

### Official Review · Reviewer_ScUX · 2024-08-01
**Learning-based Safety Filter that Leverages Simulation to Evaluate Worst-Case Scenarios for Robust Robotic Control. Question remains for scalability and sim-to-real gap.**

**Originality:** 2
**Technical Quality:** 2
**Clarity Of Presentation:** 3
**Potential Impact:** 2
**Recommendation:** 2
**Confidence:** 3

**Review:**

The paper presents an interesting idea of evaluating safety by leveraging a simulator online. The idea is straight forward and clearly presented, but the technical details and the relation to the game theory has to be clarified.

Safety-Performance Trade-off: The paper asserts that it addresses the safety-performance trade-off effectively. However, I disagree, as the hypothetical game that simulates the worst-case scenario tends to result in overly conservative control strategies. This conservative approach may hinder the robot's performance in dynamic environments. I have some concerns below.

1. Scalability Concerns: The method may not scale well to more complex scenarios, such as navigating rough terrain with significant uncertainty and unobserved disturbances. Simply adding disturbances to account for the sim-to-real gap is insufficient for robust safety.

2. Simulation and Evaluation: The use of three policies (task, fallback, adversary) during deployment to evaluate safety raises concerns about the practicality and computational overhead. Additionally, modeling the sim-to-real gap using a 6-D adversarial force  may not adequately capture real-world discrepancies, including observational errors.

3. Experimental Results: The statement in line 208, "Even for those failed trials, the gameplay filter withstands higher tugging force before it violates the safety constraints," suggests that the proposed approach may be overly conservative. It indicates that the filter does not accurately distinguish between safe and unsafe situations, potentially limiting the robot's operational capabilities.

4. Safety-Performance Trade-off: The paper asserts that it addresses the safety-performance trade-off effectively. However, I disagree, as the hypothetical game that simulates the worst-case scenario tends to result in overly conservative control strategies. This conservative approach may hinder the robot's performance in dynamic environments.

**Quality Of The Limitations Section:**

1

**Questions For Rebuttal:**

1. Disturbance Modeling: What specific types of disturbances are modeled in the experiments? The paper needs to clarify the experimental setup, including the nature and range of disturbances considered.

2. Game Theory Clarification: The connection to game theory needs to be better defined and clarified. While the paper frequently mentions game theory, it lacks a clear explanation of how game-theoretic principles are applied in the proposed method.

3. Implementation of the Simulator: Where is the simulator running during the experiments? Is it on the robot itself, or is it running on an external system? This information is crucial for understanding the feasibility and real-time performance of the proposed safety filter.

**Robotics Focus:**

4

**Summary Of Paper:**

This paper introduces a new type of safety filter designed for high-dimensional robotic systems, such as quadrupedal robots. The proposed method, known as the "gameplay filter," integrates scalable learned representations with model-based safety analysis to provide a lightweight safety assurance module for various robotic tasks. It operates by simulating "worst-case" scenarios in real-time, ensuring that the robot avoids actions leading to safety violations. The paper claims that the gameplay filter enhances safety without significantly limiting the robot's agility, and demonstrates its effectiveness through experiments on quadrupedal robots under various perturbations.

**Summary Of Recommendation:**

I recommend a major revision.

---

### Author Rebuttal · Authors · 2024-08-13

Dear Area Chair and Reviewers,

Thank you for your thoughtful reviews, feedback and suggestions. We have carefully gone through your reviews and have done our best to respond to each of your questions and concerns in detail through the Official Comments below. We have additionally updated our paper to incorporate your very insightful feedback. Please find the revised manuscript with the appendices included, as well as the supplementary video also included in the rebuttal zip folder for your convenience. Changes made in the manuscript with respect to the initial submission are highlighted in blue.

We feel that the paper has gained in clarity thanks to the review process and we truly appreciate your time and commitment. Thank you once again for your feedback and we hope you enjoy going through the revised materials. We look forward to follow-up discussions if time permits.

---

### Decision · Program_Chairs · 2024-09-04

**Decision:**

Accept

**Comment:**

The approach is interesting, but the reviewers raise several issues that should be addressed:

- Clarification is needed on the experiment setup, including the type of disturbances used
- The connection to game theory is quite unclear
- The authors should consider the effect of out-of-distribution states/actions on the method
- The novelty of the method is perhaps overstated, since much of this fits into the standard MPC framework

Please see reviewer comments for more details.

-----Post-rebuttal comments----

The rebuttal addressed several important issues brought up by the reviewers. While there is not a clear consensus among the reviewers, the overall sentiment about the paper was positive and I believe this work is above the bar for acceptance.